# Prognostic Factors in Patients with Breast Cancer Liver Metastases Undergoing Liver Resection: Systematic Review and Meta-Analysis

**DOI:** 10.3390/cancers14071691

**Published:** 2022-03-26

**Authors:** Federica Galiandro, Salvatore Agnes, Giovanni Moschetta, Armando Orlandi, George Clarke, Emilio Bria, Gianluca Franceschini, Giorgio Treglia, Francesco Giovinazzo

**Affiliations:** 1General Surgery and Liver Transplantation Unit, Fondazione Policlinico Universitario Agostino Gemelli, 00168 Rome, Italy; federica.galiandro@policlinicogemelli.it (F.G.); salvatore.agnes@policlinicogemelli.it (S.A.); dott.moschetta@gmail.com (G.M.); 2Oncology Unit, Fondazione Policlinico Universitario Agostino Gemelli, 00168 Rome, Italy; armando.orlandi@policlinicogemelli.it (A.O.); emilio.bria@policlinicogemelli.it (E.B.); 3Department of Liver Surgery, Queen Elizabeth Hospital, Birmingham B15 2GW, UK; george.clarke@uhb.nhs.uk; 4Breast Unit, Fondazione Policlinico Universitario Agostino Gemelli, 00168 Rome, Italy; gianluca.franceschini@policlinicogemelli.it; 5Academic Education, Research and Innovation Area, General Directorate, Ente Ospedaliero Cantonale, 6500 Bellinzona, Switzerland; 6Faculty of Biology and Medicine, University of Lausanne, 1015 Lausanne, Switzerland; 7Faculty of Biomedical Sciences, Università della Svizzera italiana, 6900 Lugano, Switzerland

**Keywords:** breast cancer, liver metastases, hepatic resection, prognostic factors, meta-analysis

## Abstract

**Simple Summary:**

Robust predictive and prognostic tools are needed in the management of breast cancer liver metastases (BCLMs). Until now, surgery has not been the gold standard of treatment of patients with BCLMs. The present manuscript highlights several predictive factors related to the primary tumor and the BCLM that may help to identify candidates for surgery with favorable outcomes in a large cohort of patients.

**Abstract:**

Background: The role of surgical resection of liver metastases in patients with breast cancer liver metastasis (BCLM) remains controversial. A systematic review and meta-analysis of prognostic factors related to survival after BCLM resection was performed. Methods: An electronic search of relevant publications was performed. Pooled outcome measures were expressed as hazard ratios (HRs), including 95% confidence interval values (95% CIs), and calculated through a random-effects model. Heterogeneity was tested through the I^2^ index. Results: Thirty-five publications reported analyses on prognostic factors and survival. A total of 2782 patients who underwent liver resection for BCLM were included. Positive axillary lymph nodes at breast cancer diagnosis were an unfavorable survival factor (HR 1.74, 95% CI 1.25 to 2.41, I^2^ = 0%). Cumulative predictive factor HRs (multiple liver metastases, size of the metastases, short interval between primary tumor and onset of liver disease) related to the BCLM pattern were 1.32 (95% CI 1.17 to 1.48, I^2^ = 71%) and 1.51 (95% CI 1.15 to 1.98, I^2^ = 76%) for surgical and pathological features (resection margin and presence of extrahepatic disease), respectively. Conclusion: Resection of BCLM may provide a survival benefit for selected patients. For better long-term results, surgical selection should consider both primary tumor and BCLM features such as negative axillary lymph nodes at breast resection, a single hepatic lesion, a time longer than 24 months between breast and hepatic diagnosis, and a realizable R0 liver resection. However, the high heterogeneity among studies suggests the need for an RCT to validate the present findings.

## 1. Introduction

While initially a localized disease, breast cancer may spread systemically, and the liver represents the third most common site of metastases behind the lymphatics and bone [1]. Survival in breast cancer with liver metastases (BCLM) typically does not exceed 8 months if left untreated [2]. The primary treatment for BCLM remains chemotherapy and, more recently, targeted immunotherapy. Liver metastases from other primary tumors, such as colorectal cancers, are routinely resected as part of standard management. Breast cancer rarely develops isolated liver metastases because neoplastic cells at this stage have often already reached systemic circulation with the possibility of further localization, unlike colon cancer, where the liver, through the portal system, is the first organ to be colonized. Therefore, the role of liver resection in BCLM remains controversial [3].

Several retrospective case series have been published evaluating patients’ survival following resection for BCLM, reporting 5 year survival rates ranging from 9% to over 78% [4,5,6]. These uneven results are likely due to the highly variable inclusion criteria for hepatic resection among different series, such as the presence or absence of extrahepatic disease.

On the assumption that surgical treatment of BCLM is still a debated field, it would be helpful to clarify which groups of patients may benefit from liver resection. In this regard, some reports described their results on the predictive value of factors related to both primary breast tumor and BCLM (e.g., age at diagnosis, hormone receptor status, primary tumor stage, interval between the diagnosis of the primary tumor and hepatic disease, and distribution, size, and the number of liver metastases). However, these results are not homogeneous when comparing different retrospective series.

Our study aims to perform a systematic review and meta-analysis of the current literature evidence of the prognostic 5 year overall survival (OS) factors after hepatic resection for BCLM.

## 2. Materials and Methods

The present work is reported in line with PRISMA (Preferred Reporting Items for Systematic Reviews and Meta-Analyses) and AMSTAR (Assessing the Methodological Quality of Systematic Reviews) guidelines [7].

The research protocol was registered at the International Prospective Register of Systematic Reviews (http://www.crd.york.ac.uk/PROSPERO, accessed on 30 November 2020) with the following registration number: CRD42020212007.

### 2.1. Search Strategy

An electronic search was performed using the PubMed/MEDLINE and Cochrane library databases (last search date: 27 August 2020). The following combination of terms was used: (“breast” or “non-colorectal” or “noncolorectal” or “nocolorectal” or “no-colorectal” or “non-neuroendocrine” or “noneuroendocrine” or “no-neuroendocrine”) and (“hepatic” or “liver”) and (“metastasis” or “metastases” or “metastatic”) and (“surgery” or “resection” or “surgical” or “hepatectomy”). Beyond the electronic search of bibliographic databases, the references of the selected articles were also screened manually.

### 2.2. Study Selection

Titles and abstracts of the retrieved records were independently screened by two independent reviewers (F.G. and G.M.). Studies were selected for the systematic review according to predefined criteria. Inclusion criteria were as follows: (a) original article about surgical treatment of liver metastatic breast cancer; (b) ≥10 patients resected enrolled; (c) only articles in English language; (d) survival data about resected patients. Exclusion criteria were as follows: (a) reviews, editorials, comments, and study protocols; (b) case series (less than 10 patients included) and case reports; (c) articles outside the field of interest of this review (e.g., articles focused only on results of local treatment such as radiofrequency ablation and chemoembolization); (d) articles not available in English. Duplicates were removed.

The potential overlap of patients between studies from the same hospital was evaluated. In cases of potential overlap, data were obtained from one study only, and priority was given according to criteria in the following order: (1) a study with patients that were not treated in the same time interval; (2) the study with the most significant number of patients; (3) the most recent study. Potential difficulties in the selection of these studies were solved through consensus between the reviewers.

### 2.3. Data Extraction

Two independent reviewers (F.G. and G.M.) retrieved information about the study characteristics (authors, year of publication, journal, country, study design, the time interval of the study), patient demographics, and disease features (number of resected patients, the median age at liver resection, synchronous liver metastases and/or extrahepatic disease), surgical characteristics (major/minor resections, postoperative mortality and morbidity), and survival outcomes (median follow-up, median and 3 or 5 year overall survival (OS), median and 3 or 5 year disease-free survival (DFS)).

Subsequently, studies reporting data about prognostic factors related to the primary tumor and/or liver metastases were analyzed.

Firstly, reviewers independently recorded the data to limit selection bias, and then the senior author (F.G.) screened the data.

### 2.4. Quality Assessment

The Newcastle–Ottawa Scale (NOS) [8] was used for quality assessment of the included studies.

### 2.5. Outcomes

The primary outcome was to analyze predictive factors of 5 year OS after BCLM resection.

### 2.6. Definition of Prognostic Factors

For a better understanding of the results of our analysis in the figures, we describe in detail some of the variables analyzed in the quantitative analysis:-Axillary nodes: lymph nodes retrieved at breast resection.-Multiple liver metastases: number greater than one.-Size of liver metastasis: diameter greater than 3.0 cm.-Synchronicity: hepatic diagnosis within 1 year of treatment of the primary tumor.-Age at liver diagnosis: age under or over 50 years.-Short/long interval between breast and liver diagnosis: shorter or longer than 24 months.-Extrahepatic disease: presence or not of extrahepatic disease at BCLM surgical treatment.

### 2.7. Statistical Analysis

The meta-analysis was performed using the R software suite (v3.4.0, https://www.R-project.org, accessed on 30 October 2021). The pooled outcome measure was the hazard ratio (HR) of the 5 year overall survival (OS) calculated using the random-effects model. The HR of the 5 year OS was derived from ln(HR) and standard error (SE) as previously described [9,10]. Statistical heterogeneity between trials was evaluated by χ^2^ and I^2^. Potential publication bias was investigated by funnel plot. Egger’s test was used to assess funnel plot asymmetry [11], and Macaskill’s test was used to quantify bias [12]. A *p*-value < 0.05 (two-tailed) was considered to indicate statistical significance.

## 3. Results

### 3.1. Literature Search

Results of the literature search are reported in Figure 1. Overall, 2623 studies were identified using the search strategy. A total of 2441 records were excluded after the review of the title and abstracts. Accordingly, 182 articles were included in the full-text screening; out of these, 56 publications met the inclusion criteria and were finally enrolled in the qualitative analysis, while 32 articles were included in the quantitative analysis (meta-analysis).

### 3.2. Qualitative Analysis (Systematic Review)

All included studies had a retrospective design, and only six had a control group (non-resected patients). Results on the risk of bias and reporting of quality indicators are shown in Appendix A. Only female patients were included. The age ranged between 42 and 60 years at the diagnosis of BCLM. A total of 2782 patients underwent liver resection for BCLM over a 38 year period (1980–2018). Twelve studies (21.4%) excluded synchronous liver metastases, and 13 (23.2%) included patients with extrahepatic disease at diagnosis of liver disease. Furthermore, 60% of reviewed studies reported a median follow-up of 38.2 months after hepatic resection (range: 12–81 months). Median OS was 52 months (range: 25–134.5 months) in 80% of the studies, with a cumulative 3 year OS and 5 year OS of 62.5% (range: 35.7–78%) and 42.7% (range: 9.1–78%), respectively.

Additionally, 28 out of 56 (50%) of the studies reported the disease-free survival (DFS) after hepatectomy for BCLM, with a median DFS of 25.5 months and a cumulative 3 year DFS and 5 year DFS of 29.9% (range: 8–46%) and 21% (range: 8–41.1%), respectively (Appendix A).

### 3.3. Quantitative Analysis of Prognostic Factors (Meta-Analysis)

Thirty-five publications reported an analysis on the role of prognostic factors influencing survival after hepatic resection for BCLM and were included in the meta-analysis. A publication bias was demonstrated by Egger’s test (Appendix A). The overall HR for predictive factors related to the primary tumor was 1.74 (95% CI 1.25–2.41, I^2^ = 0%) for axillary node disease at breast resection (Figure 2) and 0.90 (95% CI 0.58 to 1.41, I^2^ = 66%) for the receptor status (Figure 3). The overall HR for predictive factors related to the *BCLM pattern* was 1.32 (95% CI 1.17 to 1.48, I^2^ = 71%) (Figure 4) and 1.51 (95% CI 1.15 to 1.98, I^2^ = 76%) for surgical and pathological features, respectively (Figure 5 and Figure 6). Six comparative studies did not show any difference between the *surgical* and *systemic treatment groups* (HR 0.89, 95% CI 0.41 to 1.96, I^2^ = 90%) (Appendix A).

## 4. Discussion

BCLM correlates with a poor prognosis, and only a subgroup of patients may benefit from surgical resection as most patients are not susceptible to curative treatment [2]. No guidance has been created to select suitable patients with BCLM for hepatic resection, as a surgical option is often based on personal experience and center practice [13]. However, surgery is not routinely performed for metastatic liver tumors in patients with breast cancer. For patients with disease limited to the liver or stable extrahepatic oligometastatic disease undergoing systemic treatment, the surgical approach has been described in several retrospective series with 5 year OS ranging widely between 9% and 78% [6]. However, there is high variability in the eligibility criteria for resection, and the weakness of these criteria reflects the lack of identified prognostic, predictive factors, as previously reviewed [14,15]. Moreover, the term “oligometastatic” did not have a standardized definition in the series analyzed, as some authors considered BCLM patients with only well-controlled bone metastases [16,17]. In several studies, the term was not defined at all.

The present manuscript highlights several predictive factors related to the primary tumor and the BCLM that may help identify suitable candidates for surgery with favorable outcomes in a large cohort of patients. Currently, a limited number of comparative studies have shown mixed results comparing survival between hepatic resection plus systemic treatment and systemic treatment alone [16,17,18,19,20,21,22].

According to the experience recently reported by several authors, positive axillary nodal metastases of BC at resection of the primary tumor resulted in an adverse prognostic factor. Patients with lymphatic spread had lower survival following resection for BCLM than patients without nodal spread [18,20,23]. Positive lymph nodes may reflect a more aggressive disease with occult synchronous micro-metastases at the time of breast resection [22,24,25].

The present research shows that patients undergoing liver resection for BCLM with ER^+^ primary tumor have better survival than ER^−^ patients, defining the triple-negative status as an independent poor prognostic factor in BCLM [26,27,28]. Therefore, according to these findings, a selective approach based on receptors and lymph node status should drive the decision to resect BCLM even though the receptor status of BCLM may differ from the primary tumor [29,30]. One-third of BCLM cases are triple-negative (ER^−^/PR^−^/HER2^−^), and, in these specific cases, a surgical approach may remove the chemo-resistant BCLM. However, ER/PR positivity of BCLM does not correlate with improved survival after liver resection. Therefore, the receptor status of the primary tumor might suggest a more indolent disease [31]. Moreover, the triple-negative subtype and positive lymph nodes are unfavorable factors for BCLM resection. These variables are associated, as triple-negative BC has a higher percentage of positive lymph nodes with metastatic disease and more aggressive disease in the first years after diagnosis compared to other BC subtypes. Thus, the analysis of such a short survival subset may be biased for non-triple-negative BC patients in terms of OS and DFS.

Furthermore, other prognostic factors seem to be related to the liver metastasis properties, such as the number, size, and distribution of the liver metastases, as well as the extension and radicality of the liver resection. Patients resected for multiple BCLMs have a shorter median DFS than patients with a single BCLM [14,29,30], supporting the results of the present meta-analysis.

Decreased OS and DFS have been shown for patients with tumors ≥ 30 mm (*p* = 0.041) or >40 mm BCLM. Several authors have suggested a dimensional criterion to determinate those suitable for surgical resection of BCLM [23,24]. In our opinion, results about morphological features (e.g., size, number) should be considered with caution as they may be due to selection biases in several studies that have preferentially included patients with a single, small lesion. In addition, some authors suggested following dimensional criteria of hepatic nodules, but the evidence is not homogeneous in the literature [30,32].

Our research confirmed a short interval between breast and liver diagnosis as an adverse prognostic factor. Several studies have correlated that an interval <24 months between the diagnosis of BCLMs and the treatment of the primary tumor is a worse prognostic factor for metastatic patients undergoing liver resection [33,34,35]. Our results also support the prognostic significance of the surgical radicality for long-term outcomes, not only in terms of macroscopic residual disease, as previously described by Adam et al. in 2006 [3], but also in terms of microscopic residual disease following hepatic resection, as confirmed by a German study that indicated R1 resection as an independent prognostic factor for poor survival at multivariate analysis [35]. Radical surgery could explain why, in a favorable long-term course of disease with residual tumor or hepatic recurrence, repeat hepatectomy combined with the systemic treatment provided survival rates comparable to those after first hepatectomy [36].

It is rare for the liver to represent the only localization of metastatic breast cancer. Therefore, the majority of the case series we reviewed included patients with BCLM with extrahepatic metastases defined as a limited disease and/or well-controlled disease with a complete or partial response after the adjuvant chemotherapy, broadening the eligibility resection criteria. For example, Mariani et al. included only patients with bone metastases in their case–control study comparing surgical and nonsurgical groups for BCLM, not defining bone localization as a clear contraindication to surgery [19]. Despite this, Sakamoto et al. previously concluded that the presence of extrahepatic disease before hepatectomy was the only factor prognostic at the multivariate analysis of a poor outcome [36]. Therefore, in this context, strict tumor surveillance is necessary after treatment of the primary tumor to detect metastatic disease and select patients with isolated liver lesions that could have survival advantages from their surgical treatment [37].

Our study had several limitations due to the retrospective design of the included studies, the selection bias among inclusion criteria in the different case series, the lack of survival data about patients with BCLM resected and indiscriminately included in “non-colorectal” or “non-neuroendocrine” reports, and the prolonged time interval between the selected studies, with different surgical and medical aspects, which complicated the comparison of their results. Furthermore, we included studies over a broad time interval (1980–2018), whereby advances in medical and surgical technology [38] and management would have potentially impacted the results (e.g., HER2-targeted treatment, different imaging and pathological technologies). Moreover, the potential effect of a neoadjuvant therapy of BCLM before hepatic resection on survival was not evaluated in this meta-analysis due to a lack of data in the included studies. The present meta-analysis found a variable I^2^, reflecting the methodological heterogeneity and the limitation mentioned above. However, a meta-analysis with a time limitation would have been methodologically incorrect, as reported by PRISMA guidelines. On the other hand, the present meta-analysis is, to the best of our knowledge, the first to analyze the role of prognostic factors in both primary cancer and liver metastases for patients with BCLM and to evaluate the long-term outcomes of more than 2700 patients.

## 5. Conclusions

According to this systematic review and meta-analysis, the surgical management of BCLM remains a controversial topic. However, surgery may represent an option with survival advantages for selected patients. Nevertheless, the high heterogeneity among the studies suggests the need for an RCT to validate the present findings.

## Figures and Tables

**Figure 1 cancers-14-01691-f001:**
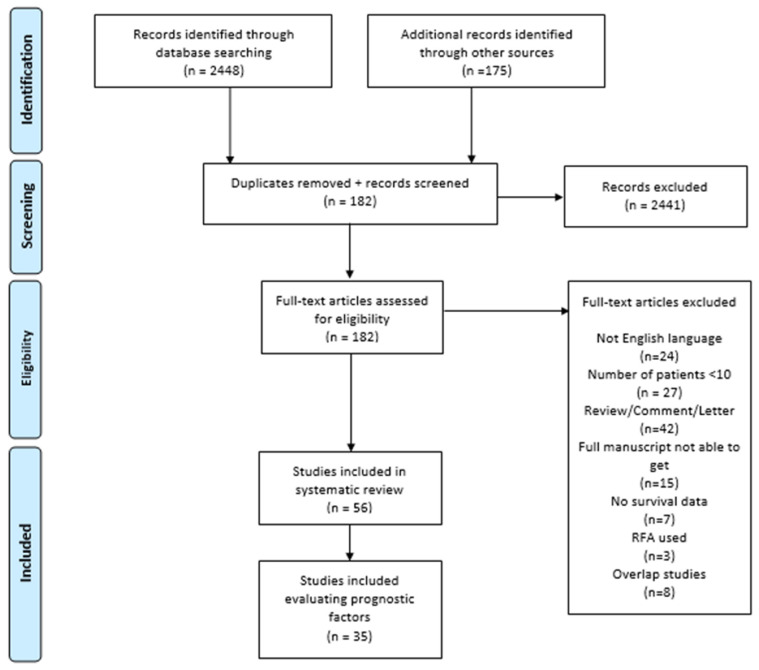
PRISMA flow diagram with included studies.

**Figure 2 cancers-14-01691-f002:**
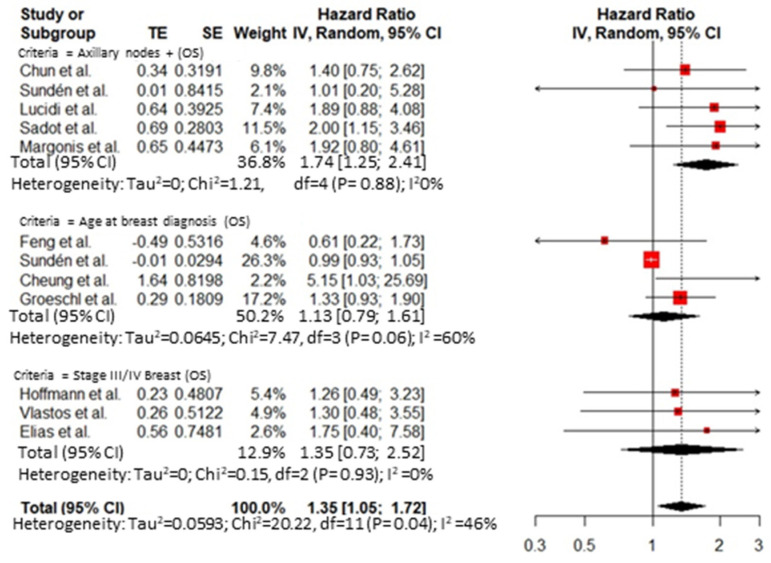
Different forest plots for predictive factors related to the primary tumor: axillary nodes; age at breast cancer diagnosis; stage III/IV breast cancer.

**Figure 3 cancers-14-01691-f003:**
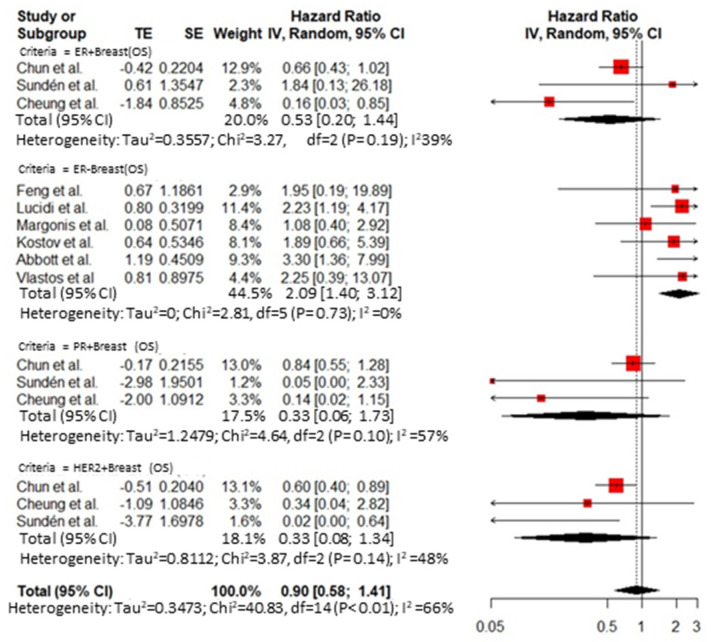
Cumulative forest plot for predictive factors related to primary tumor receptor status.

**Figure 4 cancers-14-01691-f004:**
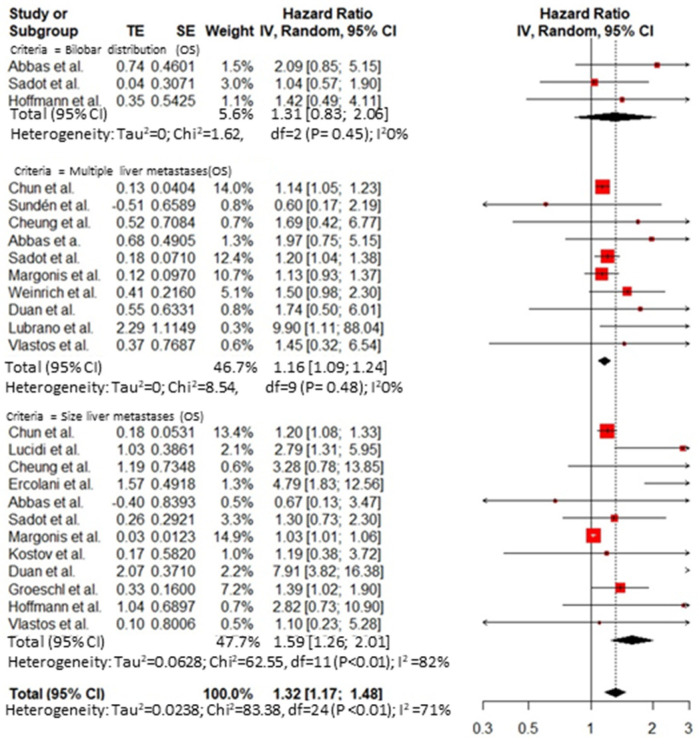
Cumulative forest plot for predictive factors related to the BCLM pattern.

**Figure 5 cancers-14-01691-f005:**
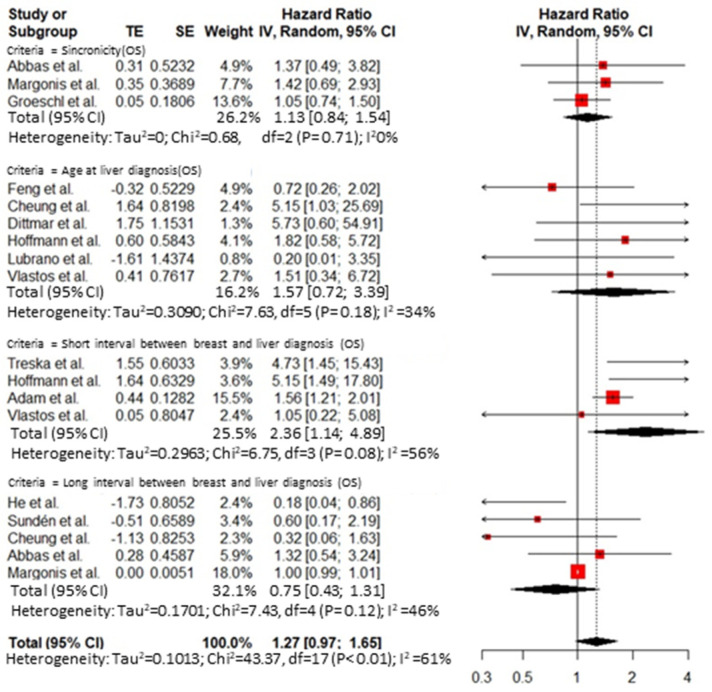
Cumulative forest plot for predictive factors related to time of presentation of the BCLM.

**Figure 6 cancers-14-01691-f006:**
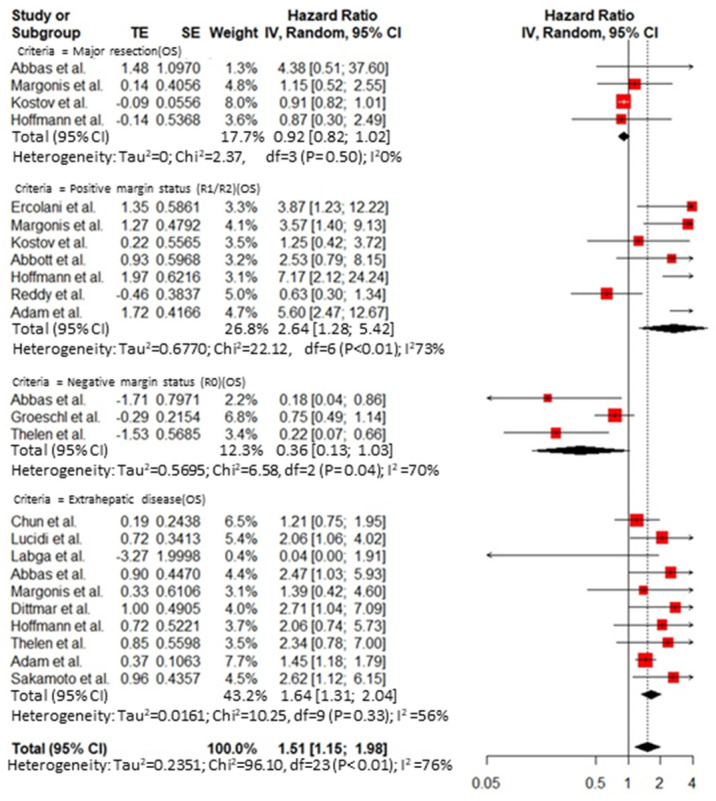
Cumulative forest plot for surgical and pathological features.

## Data Availability

The data presented in this study are available in this article (and Appendix A).

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
