# Peer review of "Prognostic Factors in Patients with Breast Cancer Liver Metastases Undergoing Liver Resection: Systematic Review and Meta-Analysis"

_cancers, 2022, doi:10.3390/cancers14071691_

Round 1

Reviewer 1 Report

Revision of manuscript titled “Prognostic Factors in Patients with Breast Cancer Liver Metastases Undergoing Liver Resection: a Systematic Review and a Meta-analysis” by Galiandro F. et al, to the journal of Cancers.

In this systematic review and meta-analysis, the authors focused on the identification of predictive and prognostic markers to determine which breast cancer patients will benefit on breast cancer liver metastasis resection (BCLM). The study is well-designed and followed recommendations of PRISMA and ARMSTAR guidelines. The statistical analyses were properly addressed. Overall, the manuscript is interesting. I encourage the authors to address the comments.

Major comments

  1. The abstract should better describe the results to explain what patients are proposed for BCLM resection as a conclusion of the study and not just leave that open.
  2. The authors found that TNBC subtype as well as positive lymph node with BC were unfavorable factors for BCLM resection. As the authors know these two variables are associated as TNBC have a higher percentage of positive lymph node with metastatic disease and more aggressive disease in the first 3/5 years after diagnosis compared to other BC subtypes. Thus, the analysis on such a short survival analysis may be bias for non-TNBC patients in terms of OS and DFS. Also, authors should discuss on how longer follow-up may impact on these results and BCLM resection. Authors need to comment in the discussion.
  3. The authors found that axillary positive BC patients may represent a prognostic factor for BCLM resection. When was axillary positive detection performed? Was that synchronous with BCLM? Or were they found at time of primary tumor resection? Please clarify and discussed.
  4. The authors are comparing patients across studies from 1980-2018. My concern is that not all the patients received the same type of surgery as the technology has significantly changed as well as imaging. For example, some patients may have non detected clinical disease due to imaging limitations. Another example is that ER+ has significantly changed over the years and furthermore not all the Pathology Dept have the same cutoff criteria. Can the authors comment about this in the Discussion?
  5. In the Introduction the authors discussed that there is a high heterogeneity in 5-years survival and that those results may depend on “…the presence or absence of extra-hepatic disease. How were those factors controlled in this meta-analysis? Please explain.
  6. I recommend to the authors to clarify if patients have previous treatment to BCLM resection. That will have a tremendous effect on DSF as well as in OS. Please explain.
  7. Why the authors discuss the HER2 therapy in the discussion. From my point of view, this is out of place and do not add to the discussion. Authors should consider removing the paragraph. “HER-2 receptors sensitise the metastatic cancer cells to targeted therapies as shown 229 by a median OS of more than 72 months in luminal B subtype (ER and/or PR+, HER2+) 230 compared to the 53-month median OS in luminal A subtype (ER and/or PR+, HER2 -) [18]. 231 Moreover, the lack of HER2 leads to more aggressive disease, and drugs such as 232 Trastuzumab are ineffective in these patients [20].”

Minor comments

  1. In Figure 2-6, gray color fonts are hard to see/read in printed version of the manuscript.
  2. In the Simple Summary, I would re-word this sentence: “Until now, surgery is not the gold standard for the treatment of patients with BCLM.”
  3. Please make sure you add a list of abbreviations.
  4. In reference 23, references’ title is in French. Please correct.
  5. Please review this sentence, “The number of liver metastases seems a significant independent factor of poor survival in resected patients [27], supporting the present meta-analysis results.”. Authors should define on whether the previous study showed or did not show that the number of liver metastasis is an independent prognostic/predictive factor for poor OS.
  6. Please labeled properly the Table as S1-3. Also, correct typos and extra spaces.
  7. Abstract, please correct. “cumulative predictive factors …related to BCLM pattern was…”.

Author Response

Thank you for your kind comments. We have modified our manuscript following the instructions received, so we list:

Major comments

  1. The abstract should better describe the results to explain what patients are proposed for BCLM resection as a conclusion of the study and not just leave that open.

R: Thank you for your comment. We added this sentence at the Conclusions of the Abstract: “For better long-term results, surgical selection should consider both primary tumour and BCLM features such as negative axillary lymph nodes at breast resection, a single hepatic lesion, a time longer than 24 months between breast and hepatic diagnosis and a realizable R0 liver resection.”.

  1. The authors found that TNBC subtype as well as positive lymph node with BC were unfavorable factors for BCLM resection. As the authors know these two variables are associated as TNBC have a higher percentage of positive lymph node with metastatic disease and more aggressive disease in the first 3/5 years after diagnosis compared to other BC subtypes. Thus, the analysis on such a short survival analysis may be bias for non-TNBC patients in terms of OS and DFS. Also, authors should discuss on how longer follow-up may impact on these results and BCLM resection. Authors need to comment in the discussion.

R: Thank you for your comment. We added this sentence at the Discussion: “Moreover, triple negative subtype as well as positive lymph node with BC are unfavorable factors for BCLM resection. These variables are associated as triple negative BC have a higher percentage of positive lymph node with metastatic disease and more aggressive disease in the first years after diagnosis compared to other BC subtypes. Thus, the analysis on such a short survival analysis may be biased for non-triple negative BC patients in terms of OS and DFS.”

  1. The authors found that axillary positive BC patients may represent a prognostic factor for BCLM resection. When was axillary positive detection performed? Was that synchronous with BCLM? Or were they found at time of primary tumor resection? Please clarify and discussed.

R: Thank you for your comment. We specified this topic in the methods adding the definition in the Paragraph “2.6. Definition of prognostic factors”: “Axillary nodes: lymph nodes retrieved at breast resection.”.

  1. The authors are comparing patients across studies from 1980-2018. My concern is that not all the patients received the same type of surgery as the technology has significantly changed as well as imaging. For example, some patients may have non detected clinical disease due to imaging limitations. Another example is that ER+ has significantly changed over the years and furthermore not all the Pathology Dept have the same cutoff criteria. Can the authors comment about this in the Discussion?

R: Thank you for your comment. We modified this topic in the Discussion about Limitations:” Furthermore, we have included studies from a broad time interval (1980-2018), probably with differences in medical (and surgical) treatment with a potential impact on the results (e.g., HER2 targeted treatment, different imaging and pathological technologies).”.

  1. In the Introduction the authors discussed that there is a high heterogeneity in 5-years survival and that those results may depend on “…the presence or absence of extra-hepatic disease. How were those factors controlled in this meta-analysis? Please explain.

R: Thank you for your comment. We modified this topic in the Discussion: “Rarely liver represents the only localization for metastatic breast cancer. Therefore, most of the case series we reviewed included patients with BCLM with extra-hepatic metastases defined as a limited disease and/or well-controlled disease with a complete or partial response after the adjuvant chemotherapy broadening the eligibility resection criteria. For example, Mariani et al. included only patients with bone metastases in their case-control study comparing surgical and no surgical groups for BCLM, not defining bone localization as a clear contraindication to surgery [19]. Despite this, Sakamoto et al. previously concluded that the presence of extra-hepatic disease before hepatectomy was the only factor prognostic at the multivariate analysis of a poor outcome [36]. Therefore, in this context, strict tumour surveillance is necessary after treatment of the primary tumour to detect metastatic disease and select upfront patients with isolated liver lesions that could have survival advantages from their surgical treatment [37].”.

  1. I recommend to the authors to clarify if patients have previous treatment to BCLM resection. That will have a tremendous effect on DSF as well as in OS. Please explain.

R: Thank you for your comment. We specified this topic in the Discussion about limitations:” Moreover, the potential effect of a neoadjuvant therapy of BCLM before hepatic resection on survival was not evaluated in this meta-analysis due to lack of data about this parameter among the included studies.”.

  1. Why the authors discuss the HER2 therapy in the discussion. From my point of view, this is out of place and do not add to the discussion. Authors should consider removing the paragraph. “HER-2 receptors sensitize the metastatic cancer cells to targeted therapies as shown 229 by a median OS of more than 72 months in luminal B subtype (ER and/or PR+, HER2+) 230 compared to the 53-month median OS in luminal A subtype (ER and/or PR+, HER2 -) [18]. 231 Moreover, the lack of HER2 leads to more aggressive disease, and drugs such as 232 Trastuzumab are ineffective in these patients [20].”

R: Thank you for your comment. As suggested, we removed this sentence.

Minor comments

  1. In Figure 2-6, gray color fonts are hard to see/read in printed version of the manuscript.

R: Thank you for your comment. We modified Figures 2-6 with better (Black) color fonts.

  1. In the Simple Summary, I would re-word this sentence: “Until now, surgery is not the gold standard for the treatment of patients with BCLM.”

R: Thank you for your comment. We modified this sentence in the Simple Summary: “Until now, there are not standardized prognostic tools to select BCLM patients for surgical resection.”.

  1. Please make sure you add a list of abbreviations.

R: Thank you for your comment. We added a list of abbreviations at the end of the Manuscript.

  1. In reference 23, references’ title is in French. Please correct.

R: Thank you for your comment. We have translated the title into English.

  1. Please review this sentence, “The number of liver metastases seems a significant independent factor of poor survival in resected patients [27], supporting the present meta-analysis results.”. Authors should define on whether the previous study showed or did not show that the number of liver metastasis is an independent prognostic/predictive factor for poor OS.

R: Thank you for your comment. We removed this sentence and this wrong reference. Sorry.

  1. Please labeled properly the Table as S1-3. Also, correct typos and extra spaces.

R: Thank you for your comment. As suggested, we modified Table S1-3.

Reviewer 2 Report

Overall, this is a very interesting systematic review and meta-analysis, with important findings for the long-term treatment and management of breast cancer patients with liver mets. There are a few issues with the written English, and I suggest making the corrections as outlined in the attached document, however, methodologically, this study is sound, and is suitable for publication in this journal.

Author Response

Thank you very much for your comments. In the re-submitted form, the English has been reviewed by one of the co-authors (GC), an English native speaker.